# Dizziness in Fabry Disease

**DOI:** 10.3390/biomedicines13020249

**Published:** 2025-01-21

**Authors:** Aslak Broby Johansen, Ulla Feldt-Rasmussen, Mads Klokker

**Affiliations:** 1Copenhagen Hearing and Balance Center, Department of Otorhinolaryngology, Head and Neck Surgery and Audiology, 2100 Copenhagen, Denmark; mads.klokker@regionh.dk; 2Department of Nephrology and Endocrinology, Rigshospitalet, 2100 Copenhagen, Denmark; ufeldt@rh.dk; 3Institute of Clinical Medicine, Faculty of Health and Medical Sciences, Copenhagen University, 2200 Copenhagen, Denmark

**Keywords:** Fabry, dizziness, balance issue, vertigo, dizziness handicap index, oto-neurological examination, polypharmacy

## Abstract

**Background/Objectives**: Fabry disease is an X-linked lysosomal storage disease. Earlier studies have mentioned dizziness/balance issues and vestibular involvement as a symptom of Fabry disease. Research on the matter remains scarce. This pilot study aims to show the prevalence of dizziness/balance issues and whether it is due to peripheral, central, or other factors. **Methods**: A Dizziness Handicap Inventory, with added questions, was sent out to 91 Fabry patients to estimate the prevalence of dizziness/balance issues. Additionally, this study reports analyses from eight Fabry patients with self-reported dizziness/balance issues who were offered referrals for in-depth investigations of their condition. All eight underwent a comprehensive oto-neurological examination, Videonystagmography, a Video Head impulse test, vestibular myogenic evoked potential, and audiometry. **Results**: A total of 55 of the 91 patients with Fabry disease answered the survey. Of these, 78.2% felt symptoms of dizziness/balance issues. The most common form of dizziness/balance issues was short-lasting attacks. All eight ENT-examined patients had normal outer and middle ear conditions. Five of eight Fabry patients had abnormal results in the optokinetic test and audiometry. **Conclusions**: The survey showed a high prevalence of dizziness/balance issues in Fabry patients. The abnormal optokinetic test suggested a central cause and was the only objective measurement we found that could lead to an explanation for dizziness/balance issues. Polypharmacy was present in all eight examined patients and could also explain the dizziness/balance issues in Fabry patients. There is no other clear pattern regarding the characteristics of dizziness/balance issues in Fabry patients in this exploratory study.

## 1. Introduction

Fabry disease (FD)—also called Anderson–Fabry disease—is an X-linked, lysosomal deposition disease. An error in the gene encoding alpha-galactosidase A (*GLA*-gene) causes FD. The deficiency of the enzyme alpha-galactosidase A leads to an accumulation of glycosphingolipids [1,2]. Nearly 1000 different mutations, with varying pathogenicity, in the *GLA*-gene have been found [1,3]. Lysosomal deposits of glycosphingolipids lead to various pathological processes, such as cell death, oxidative stress, damage to small vessels, ion channel defects in the endothelium, tissue ischemia, and fibrosis [4].

Symptoms include chronic pain, kidney, and heart problems, paresthesia, tinnitus, hearing loss, and dizziness. Furthermore, Fabry crisis—episodes of severe pain in the extremities that radiate proximally—can occur [1,2].

The disease affects all ethnic groups. The rarity of FD makes an accurate prevalence challenging to obtain. Reported prevalence ranges from 1:476,000 to 1:40,000, or even more frequent [2,4,5,6].

FD has numerous phenotypes, which vary from early-onset severe courses to milder cases with later onset [2,4,5]. Several studies have described the renal or cardiac variant according to the location of the predominant organ affection [4,7,8]. For untreated FD cases, the average life expectancy is reduced by 15 years for females and 20 years for males. The most common causes of death are due to kidney or heart disease—with a predominance of the latter due to treatment possibilities [9]. Treatment with enzyme replacement (ERT) or enzyme stabilization may reduce specific symptoms experienced by FD patients to reduce the risk of heart disease [10,11,12].

Other studies have mentioned dizziness as a symptom of FD [13,14,15,16,17,18,19]. However, research in this area is scarce without suggested causes of symptoms. Thus, this pilot project investigated the overall incidence of dizziness/balance issues and screened for possible causes in a sub-population of the dizziest Danish patients with FD and their level of discomfort.

## 2. Materials and Methods

We performed a survey of Danish FD patients and their experience of dizziness/balance issues. Furthermore, a selection of patients afflicted by dizziness/balance issues underwent investigations of these problems. All patients in this study were affiliated with the Department of Hormone and Metabolic Diseases at Rigshospitalet, where there are 95 FD patients, with a ratio of 2:1, females and males, respectively. The patient cohort has been described in previous publications [20,21].

Vestibular examinations were performed at Copenhagen Hearing and Balance Center (CHBC) at the Department of Otorhinolaryngology, Head and Neck Surgery and Audiology at Rigshospitalet, Denmark.

### 2.1. Survey

A version of the Dizziness Handicap Inventory (DHI), with added questions, was sent to 91 patients with whom the department has contact. The recruitment period lasted from August 2021 to November 2021. The added questions are seen in Table 1. The 25 questions of the DHI, in Danish, were not modified. The DHI assessment was according to the standard scoring principles of the DHI [22].

### 2.2. Investigations

FD patients, who complained that dizziness/balance issues were a great nuisance for them, were offered a referral for a routine, albeit exhaustive, vestibular investigation at CHBC—of whom eight accepted the referral. These eight FD patients comprised the pilot group of this study.

Examinations were conducted from September 2021 to November 2021 and included in the patient’s medical records. Medical records were accessed on the same day of examination, and data from the investigation were included in the dataset right away. We examined patients with a general ear–nose–throat examination focusing on neuro-otological factors. Bedside vestibular examinations were conducted (i.e., Head shake test, Unterberger (Fukuka, Japan) step test, and Subjective Visual Vertical/Horizontal assessment (SVV/SVH) (performed by the “Bucket Test”)) [23,24,25]. A rough sensitivity examination of the lower extremities (UEs) was performed using a 20-degree Celsius unheated metal stick versus the examiner’s warm index finger [26].

The same specialist doctor and vestibulogist assistant examined all patients at CHBC.

During videonystagmography (VNG) (FireWire Glasses from VisualEyes™, Interacoustics, 5500 Middelfart, Denmark), we performed a saccade examination, smooth pursuit, Dix–Hallpike, optokinetic test, caloric test, and test for spontaneous nystagmus [23]. Using a repositioning chair (TRV Chair, Interacoustics, 5500 Middelfart, Denmark), we tested for Benign Positional Paroxysmal Vertigo (BPPV). A Video Head Impulse Test (vHIT) (EyeSeeCam from VisualEyes™, Interacoustics, 5500 Middelfart, Denmark) was used to examine the vestibular-ocular reflex (VOR).

Vestibular-evoked myogenic potential (VEMP) was examined by Ocular-VEMP (oVEMP) and Cervical-VEMP (cVEMP) [27].

Pure tone audiometry was performed binaurally for frequencies between 250 Hz and 8000 Hz and tested for air and bone conduction. Thresholds for the stapedius reflex were measured by ipsi- and contralateral stimulation, while masking with noise was used where necessary. We considered the normal range for hearing to be 20 dB or better.

The discrimination score was determined with words and numbers read to the patient at a volume above the hearing threshold. Furthermore, Weber’s test and tympanometry were conducted [28,29].

### 2.3. Statistics

Fisher’s exact test was used with contingency tables to determine the statistical significance. Statistical significance was set to 5%. Confidence intervals (CIs) of 95% were calculated for the DHI scores. Age groups were comparable to those used by Köping et al., 2018 [18].

## 3. Results

### 3.1. Survey

The majority of the Danish FD patients answered the survey (response rate of 60.4%). Twice as many females as males responded to the survey, which correlated with the sex distribution of the entire disease population according to the X-linked inheritance and indicated unbiased recruitment for the survey study [20,21].

As seen in Table 1, 78.2% stated that dizziness/balance issues were a symptom. Over half of the patients reported that they experienced symptoms weekly or more frequently. The most common form of dizziness/balance issues was attacks of shorter duration.

Table 2 shows scores from the DHI divided by whether the question is related to emotional, physical, or functional factors.

No significant association between sex/age and dizziness/balance issues was found.

### 3.2. Investigations

In this pilot project, we looked at results from investigations for dizziness/balance issues in eight FD patients—six females and two males. The mean age of all patients examined was 51.5 years (Standard deviation (SD) 22.04 years). The examined patients all suffered from dizziness/balance issues, and a summary of the DHI scores in this pilot group is as follows: mean = 48.50, SD = 19.94, and 95% CI: 34.68–62.32. There was no general tendency for the problem to be of nautical or rotational nature. All examined patients had normal outer and middle ear conditions, as seen by otomicroscopy and tympanometry.

vHIT (normal for 6/7), Head Shake test (normal for 5/8), Unterberger (Fukuka, Japan) step (normal for 6/8), and Romberg’s test (normal for 6/7) were normal for most patients.

VNG showed that no FD patients had spontaneous nystagmus, and all had normal results in the saccadic test and Dix–Hallpike. The optokinetic test was abnormal in five of eight patients, while smooth pursuit was only abnormal for a single patient. During the VNG examination, the wireless glasses had a technical issue, so only the left eye could be examined.

Caloric testing was normal for four out of six patients but could not be performed on two patients as they would or could not participate in the examination.

cVEMP was abnormal for half of those studied but was normal in four of five patients under 60. oVEMP examinations showed no safe potentials, which are attributed to faults in the equipment.

Patient records showed that none of the eight had bothersome tinnitus when asked.

All eight were on more than five medications shown in their medication list. The list of prescribed medications that the FD patients took is shown in Appendix A.

## 4. Discussion

This pilot study investigated if dizziness/balance issues are a problem for FD patients and whether peripheral, central, or other factors cause it.

Nearly four in five FD patients reported suffering from dizziness/balance issues. More females than males experienced dizziness/balance issues. In the Western population, up to 35% experience dizziness during their lifetime [30]. Dizziness/balance issues affects respondents more than the background population. This study compared the point prevalence in FD patients with lifetime risk in the general population. Epidemiological data on dizziness/balance issue prevalence are limited. Data from non-Danish countries were utilized, assuming comparability to Danish conditions.

It is noted that physical factors of dizziness/balance issues have the biggest impact on the level of handicap for FD patients and are significantly higher than emotional factors, but no other significant differences were found between the groups.

Dizziness/balance issues in the form of attacks of shorter duration indicated a peripheral lesion such as BPPV or Ménière’s disease, as centrally caused dizziness is often prolonged and diffuse. This is emphasized by the physical aspects of the DHI responses since positional changes can trigger BPPV [29].

The eight examined patients all suffered from dizziness/balance issues and had statistically significant higher total DHI scores compared with the rest of the survey. However, caloric testing indicated no peripheral vestibular involvement for most [31]. However, this could be incidental because of the sample size. In contrast, Eyermann et al. 2019 show that 56% of examined patients show lateral semicircular canal deficits with VNG and caloric testing [16]. Keilmann et al. 2006 showed that uni- and bilateral vestibular deficits were present in FD. However, they attributed orthostatic reasons as the leading cause of dizziness [17]. In comparison, Asquier-Khati et al. 2022 refer to a theory that cochleovestibular ischemia is the reason for the involvement of the vestibular system [32].

Several of the vestibular tests used are inaccurate by themselves but can substantiate each other. Therefore, the results of these examinations do not indicate that peripheral vestibular lesions are generally present in FD patients [24,25,28,29,33]. None of the examined patients showed evidence of BPPV during the Dix–Hallpike maneuver, emphasizing that peripheral causes were not apparent [34,35].

cVEMP was largely normal for FD patients under 60. According to Papathanasiou et al. 2014, age over 60 makes it difficult to demonstrate safe potential. Thus, this study provided no evidence that the sacculus and inferior vestibular nerve functions are generally impaired [27]. Conversely, Carmona et al. 2017 previously showed that 45 % of FD patients had abnormal VEMP examinations, suggesting an involvement of the inferior vestibular nerves [19].

The optokinetic test results point to a central lesion for several patients with FD [28,36]. Conversely, saccadic, and smooth pursuit examinations showed no abnormalities for almost everyone. Spontaneous nystagmus was not present in any of the patients studied. In addition, SVV/SVH was normal. These tests may indicate both central and peripheral causes of vertigo, depending on the direction of nystagmus and the outcome of SVH/SVV [23,37,38].

Audiometry showed sensorineural hearing loss in most of those examined. There was no lateralization in Weber’s test, which emphasizes no sign of conductive hearing loss. The discrimination score (DS) was normal for everyone, indicating no retrocochlear hearing loss [28,29]. This is in line with other studies that have shown sensorineural hearing loss in FD patients—even in the same cohort [18,19,20,32]. However, there are studies indicating sensorineural hearing loss with retrocochlear involvement, both within the examined group and compared to healthy relatives [15,32]. Conversely, tinnitus was not observed among the examined patients that have otherwise been present in patients in the above-mentioned studies.

We attempted to find whether the dizziness/balance issues in FD patients was of central or peripheral causes. Our results, and when compared to other studies, complicate finding a clearcut central or peripheral vestibular cause for dizziness/balance issues. A possibility could be a combination of these, but this would require further studies to clarify.

Polypharmacy and other afflicted organ systems can also contribute to dizziness/balance issues [17,29]. Several define polypharmacy as being on five or more medicines [39]. All eight patients studied had more than five medications on their medication list. Furthermore, several medications shown in Appendix A, and especially the ones used in the ERT/enzyme stabilizing therapy of FD in Denmark at the time of examination, list dizziness as a common side effect.

Another possible cause of dizziness/balance issues is that the sensitivity study in UE was abnormal in three out of six cases. Skin sensitivity contributes to the sense of position (proprioception), and affliction could cause instability [33,35].

A strength of this study is that the majority (55/95 (57.9%)) of Danish FD patients answered the survey, as Rigshospitalet is Denmark’s only formal affiliation for FD patients. We have used validated methods for the survey, and the survey is in an internationally recognized format [40]. The examinations are a part of the conventional investigation for dizziness/balance issues at the CHBC at Rigshospitalet.

An important limitation of this study is, since Fabry disease is rare, few individuals were examined. Therefore, whether a result is a trend, or an incidental finding is uncertain. Cardiac (such as arrhythmias), neurological (such as stroke), and renal pathologies as described by Eng et al. 2006 [2] are risk factors for dizziness. These problems are, as mentioned, associated with FD. For the analyses of this survey and the investigations, no adjustments or further examinations (such as a complete neurological examination and/or medical imaging) were made for other pathologies, and, therefore, there is a risk of confounding. Thus, it would be relevant to investigate more patients and better be able to adjust for possible confounders and effect modifiers. In addition, phenotype characterization with data from other symptoms, treatment, disease onset, and biochemical variables (such as accumulated waste products) can be correlated to dizziness/balance issue symptoms.

Selection bias may have influenced the study results. Patients for whom dizziness/balance issues was a significant nuisance may have been more likely to respond. Furthermore, we only analyzed the results of investigations for patients who specifically complained of dizziness/balance issues. Non-response bias may show a more significant proportion of patients with dizziness/balance issues in the sample than in the entire FD population.

In surveys, there is a risk of response bias, as the patient can describe the condition as worse or better than it is.

It has not been possible to find other studies in which FD patients have had to respond to the DHI. Future studies may further investigate whether dizziness/balance issues affect the quality of life using the DHI.

In conclusion of this pilot project, the prevalence of dizziness/balance issues in Danish FD patients is high. A total of 78.2% have reported dizziness/balance issues, which is considerably more than the Western background population (35%).

The survey shows that dizziness/balance issues affect a large proportion of patients. It affects their daily lives and, to some extent, hinders their daily activities. Respondents have reported dizziness/balance issues as a frequent problem that can occur weekly or even more frequently. It is difficult to describe how FD patients experience dizziness/balance issues accurately. Common to FD patients, dizziness/balance issues most often manifests itself in attacks in this study.

Our investigation of the eight FD patients with dizziness/balance issues showed a probable central cause of dizziness/balance issues due to an abnormal response to the optokinetic test. The optokinetic test is the most likely objective measure to indicate a cause of dizziness/balance issues in FD patients. However, a mixed peripheral and central cause of dizziness/balance issues in FD patients cannot be excluded, nor can other causes, such as polypharmacy.

## Figures and Tables

**Table 1 biomedicines-13-00249-t001:** Characteristics of dizziness/balance issues among Fabry patients.

		Female	Male	Total	Percent
**Sex**		40	15	55	100.0
**Age—Average**	Years	47.7 (18.5)	41.0 (12.3)	46.2 (17.2)	-
**Do you have any form of dizziness/balance issues?**	Yes	34	9	43	78.2
	No	6	6	12	21.8
**How often do you experience dizziness/balance issues? I notice symptoms:**	Daily/constant	14	1	15	27.3
	Weekly	10	4	14	25.5
	Monthly	9	3	12	21.8
	A few times a year	5	2	7	12.7
**How long do your dizziness/balance issues last?**	Seconds	15	6	21	38.2
	Minutes	22	3	25	45.5
	Hours	7	1	8	14.5
	Days	0	0	0	0.0
	Constant	2	0	2	3.6
**How do your symptoms manifest?**	In attacks	23	6	29	52.7
	Constant with attacks	0	0	0	0.0
	Constant	3	1	4	7.3
	Accumulated attacks with symptom-free periods	6	1	7	12.7
**How bothered are you by your dizziness/balance issues?—Average**	0–100	36.3 (24.5)	32.8 (26.5)	34.5 (24.6)	-

Standard deviations (SDs) are noted in parenthesis next to numerical parameters.

**Table 2 biomedicines-13-00249-t002:** Dizziness Handicap Inventory (DHI) score.

Category (Max Point)	Emotional (36 p)	Physical (28 p)	Functional (36 p)	Total (100 p)
**Mean**	7.02	12.14	10.42	28.98
**SD**	6.49	6.43	9.95	20.32
**95% CI**	5.30–8.74	10.44–13.84	7.79–13.05	23.61–34.35
**Quartile 1**	2	7	2	15
**Quartile 2 (Median)**	4	12	8	28
**Quartile 3**	11	16	15	38

## Data Availability

The data used in this study have not been used in other studies. The datasets generated and analyzed during the current study are not publicly available due to data privacy reasons but are available from the corresponding author upon reasonable request.

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
