# Peer review of "Dizziness in Fabry Disease"

_biomedicines, 2025, doi:10.3390/biomedicines13020249_

Round 1
Reviewer 1 Report
Comments and Suggestions for Authors
As you report in the text thera are a lot of limitations in the study. The small number of participants, the limited design (as consequence of the number of examined people), the statistical analysis (the relation between the group and the parameters) have to be improved. In any case, the authors should stress that it is a pilot-study.
Author Response
Comments 1:
As you report in the text thera are a lot of limitations in the study. The small number of participants, the limited design (as consequence of the number of examined people), the statistical analysis (the relation between the group and the parameters) have to be improved. In any case, the authors should stress that it is a pilot-study
Response 1:
Thank you for your valuable feedback. The revised manuscript has emphasized that this is a pilot study and highlight these aspects more clearly to provide appropriate context for the findings - highlight are made throughout the manuscript, but please refer especially to the abstract, end of introduction, the method section and the conclusion. The study limitations are also further revised and expanded. In regard to statistics, we have included 95% CI for the DHI scores, in order to improve the comparability between the findings - please refer to table 2. The changes are highlighted in Word - an easy way to read the text is to go to the Review-tab and select Simple markup under Tracking - the full changes are seen with All markup. Thank you for taking the time to review our work and provide important input - please elaborate if ther are other revisions to out manuscript.Reviewer 2 Report
Comments and Suggestions for Authors
It is interesting and clearly presented article about vertigo in patients with Fabry disease.
I have few comments;
- as there are often cerebrovascular events in FD patients - were these patients assessed neurologically, Was brain MRI performed?
- further could dizziness be a part of heart disease (arrhytmia?)
Not only otoloaryngological cause may lead to dizziness to it woul be great to mention about it as well as strokes and heart arrythmias in FD patients in study limitations or introduction
Author Response
Comments 1:
It is interesting and clearly presented article about vertigo in patients with Fabry disease.
I have few comments;
- as there are often cerebrovascular events in FD patients - were these patients assessed neurologically, Was brain MRI performed?
- further could dizziness be a part of heart disease (arrhytmia?)
Not only otoloaryngological cause may lead to dizziness to it woul be great to mention about it as well as strokes and heart arrythmias in FD patients in study limitations or introduction
Response 1:
Thank you for your thoughtful and constructive comments on our article. We are pleased to hear that you found it interesting and clearly presented. In regard to the need to expand on study limitations. In addition to otolaryngological causes, it is essential to acknowledge the potential contributions of strokes, arrhythmias, and other systemic factors that can lead to dizziness in FD patients. We have revised our manuscript accordingly - we refer to lines 200-250 (approximately) for the changes we have made. The changes are highlighted in Word - an easy way to read the text is to go to the Review-tab and select Simple markup under Tracking - the full changes are seen with All markup.Reviewer 3 Report
Comments and Suggestions for Authors
The manuscript delves into the epidemiology of dizziness in patients with Fabry disease, examined the results of a transversal survey sent to patients and some of the subjects were thoroughly evaluated in person.
It is an important manuscript that may trigger additional studies in this field.
Please see below for my questions and comments.
Abstract
1. Based on my comments, please rewrite the abstract to include information on the method of selection of the 8 patients evaluated in person.
2. Please consider adjusting the wording to include imbalance or unsteadiness alongside dizziness.
Introduction
3. Please use italics when referring to genes (GLA)
Methods
4. It is not clear to me how the 8 FD patients with dizziness were selected for further analysis. Please clarify. Based on the results presented later in the manuscript, patients had mild symptoms according to the scoring system of the DHI questionnaire. How were the “more affected" patients selected? Please also define the criteria for selection of more affected patients.
5. Authors use “a selection of vestibular afflicted patients”, but later inform that most likely the cause of the symptoms may have been from central origin. Please clarify.
6. Author used a “modified” version of the DHI questionnaire. Please inform the modifications, cite the references or include as Appendix a version of the modified questionnaire.
7. Please revise “primo August”
Results
8. Imbalance may not necessarily be related to dizziness. Authors mention that 78.2% of the participants were considered to have dizziness. Knowing that FD patients may have peripheral and central nervous system affection, how did authors exclude neurological problems in those who were not evaluated in person? I understand that only the 8 patients evaluated in person had their medical records reviewed.
9. Were do the questions presented in Table 1 come from? Are they part of the modification in the DHI? Please clarify and, if adequate, cite the reference of these questions. If they were specifically made for this study, why include “imbalance” in the question?
10. Again, authors should be more specific as to how “the dizzier patients” were selected – please inform DHI scores, or any relevant information to better inform the reader.
Conclusion
11. Authors chose to conclude that dizziness AND UNSTEADINESS prevalence is high. Please consider using both throughout the manuscript since the results are more clearly related to both rather than to dizziness alone.
Author Response
Thank you so much for your thoughtful comments and for taking such a structured approach—it has been of great help in improving the manuscript. We truly appreciate your insights!
We have addressed your feedback, and you’ll find responses to your comments in the attached file.

Round 2
Reviewer 1 Report
Comments and Suggestions for Authors
It is better now. I don't have any recommedations.
Author Response
Comments 1: It is better now. I don't have any recommedations.
Response 1: Thank you for your response. We are glad to hear that things are better now. We appreciate your feedback.
Reviewer 3 Report
Comments and Suggestions for Authors
Authors addressed all my comments and suggestions. No further comments.